# Operational Characteristics of Immobilized *Ochrobactrum* sp. CUST210-1 Biosystem and Immobilized Chromate Reductase Biosystem in Continuously Treating Actual Chromium-Containing Wastewater

**Guey-Horng Wang** [1], **Teh-Hua Tsai** [2], **Ching-Hung Chiu** [3], **Chiu-Yu Cheng** [3] **and Ying-Chien Chung** [3,*]

[1] Research Center of Natural Cosmeceuticals Engineering, Xiamen Medical College, Xiamen 361008, China; wanggh@livemail.tw

[2] Department of Chemical Engineering and Biotechnology, National Taipei University of Technology, Taipei 10608, Taiwan; thtsai@ntut.edu.tw

[3] Department of Biological Science and Technology, China University of Science and Technology, Taipei 11581, Taiwan; chiuchinghung3@gmail.com (C.-H.C.); cycheng@cc.cust.edu.tw (C.-Y.C.)

[*] Correspondence: ycchung@cc.cust.edu.tw; Tel.: +886-2278-1862; Fax: +886-2786-5456

**Abstract:** Cr(VI) detoxification by biotreatment is considered one of the most practical detoxification methods, especially at low-to-medium concentrations. Although the capabilities of chromium-reducing bacteria and related enzymes in removing Cr(VI) have been explored, little is known about their differences in engineering applications. In this study, *Ochrobactrum* sp. CUST210-1 was isolated and its chromate reductase identified and separated as biological elements in biosystems developed for Cr(VI) removal. Results indicate that intracellular $Cr(OH)_{3(s)}$ accounted for 88.01% of Cr(VI) reduction product, and a possible reduction mechanism was exposed. The chromate reductase in *Ochrobactrum* sp. CUST210-1 was ChrR protein, and its crystal structure was revealed. The toxicity of Cr(VI)-containing wastewater was decreased by 57.8% and 67.0% (at minimum) by the CUST210-1 strain and ChrR, respectively. The *Ochrobactrum* sp. CUST210-1 biosystem demonstrated good adaptability to pH (7–9), and the ChrR biosystem exhibited high removal efficiency (>98.2%) at a wide range of temperatures (25 °C–40 °C). The outlet Cr(VI) concentration of the CUST210-1 biosystem met the industrial discharge limit of 0.5 mg $L^{-1}$ when the inlet Cr(VI) concentration in the actual Cr(VI)-containing wastewater was <430 mg $L^{-1}$. The stricter water quality standard of 0.05 mg $L^{-1}$ could be complied with by the immobilized ChrR biosystem when <150 mg $L^{-1}$ Cr(VI) concentration was introduced. These developed biosystems can be used in the bioremediation of various Cr(VI)-contaminated wastewaters. Regarding capital costs, those of the CUST210-1 biosystem were higher. To our knowledge, this is the first report comparing differences in the economic and operational characteristics of bacteria and enzyme biosystems for Cr(VI) removal.

**Keywords:** bioremediation; chromium-reducing bacteria; chromate reductase; hexavalent chromium

## 1. Introduction

In the environment, the two most stable oxidation states of chromium are its hexavalent (Cr(VI)) and trivalent (Cr(III)) forms [1]. Cr(III) is considered less toxic than Cr(VI), because it is conventionally considered an essential micronutrient in animal and human diets. By contrast, Cr(VI) is highly toxic, mutagenic, carcinogenic, and teratogenic to biological systems and is readily taken up by

organisms via the sulfate transport system [2]; hence, most regulatory agencies worldwide consider Cr(VI) a priority pollutant [3,4]. Chromium compounds are used in chromium electroplating, metal processing, leather tanning, metal corrosion inhibition, papermaking, pigment manufacturing, and other industrial applications [5]. Cr(VI)-containing industrial effluents without proper treatment are released, intentionally or accidentally, into environments in developing and underdeveloped countries [6]; thus, the contamination of surface waters and groundwaters with Cr(VI) is a potential concern. Concentrations of Cr(VI) in these effluents must be reduced to permissible limits by using the appropriate technology before the effluents are discharged into the environment. The Environmental Protection Agency in Taiwan has set the maximum contaminant levels for Cr(VI) in most industrial effluents at 0.5 mg $L^{-1}$ and for fisheries water, drinking water, groundwater, and surface water at 0.05 mg $L^{-1}$.

Many physical and chemical methods have been developed for Cr(VI) removal, such as precipitation, adsorption, ion exchange, electrochemical treatment, evaporation, and reverse osmosis [5]. Nevertheless, existing methods are inefficient and not economically feasible, especially when Cr(VI) contamination in wastewater must be reduced to a low level to meet strict environmental standards, such as 0.5 mg $L^{-1}$ [7]. Several researchers have reported that the biological detoxification of Cr(VI) by bacteria and their enzymes is considered one of the most practical removal methods, especially at low-to-medium concentrations of Cr(VI), due to its ecological compatibility and low cost [4,8].

Bacteria endowed with the capacity to reduce Cr(VI) levels are termed chromium-reducing bacteria (CRB) [9]. Numerous varieties of CRB reduce Cr(VI) to Cr(III), including *Bacillus cereus* [10], *Ochrobactrum* sp. [11], *Paenibacillus ferrarius* [12], *Exiguobacterium aestuarii* [13], *Shewanella* sp., *Desulfovibrio* sp., *Enterobacter* sp., *Micrococcus* sp., *Pseudomonas* sp. [8], and several other species [5]. Although most of these CRB have been isolated from various Cr(VI)-contaminated sites, the availability of CRB is an essential prerequisite to meeting strict environmental standards, regardless of their removal efficiency or capacity. Thus, the feasibility of chromate reductase in removing Cr(VI) is considered, because it functions without the requirement of high organics concentration and complicated biochemistry processes like bacterial cells but possesses a high substrate specificity for Cr(VI) reduction.

Cr(VI) reduction or detoxification by CRB is regarded as an enzyme catalysis reaction attributed to soluble chromate reductase or cell membrane-bound chromate reductase [14]. Cell immobilization techniques are typically employed in wastewater treatments, because they result in a solid–liquid separation that is more stable, and thus, the products are easier to reuse [15]. Similarly, the use of immobilized enzymes instead of living bacterial cells may overcome the limit of the toxicity of Cr(VI) to the chromate reductase and produce dilute Cr(VI) residue, in addition to achieving the aforementioned advantages [16]. Chromate reductase is divided into four categories: (1) cytochrome C, (2) flavin protein, (3) old yellow enzymes, and (4) hydrogenases [7]. Among chromate reductase enzymes, soluble chromate reductase such as ChrR, nfsA, and yieF are suitable for development as biocatalysts for Cr(VI) bioremediation, because they are more amenable to protein engineering to suit the environmental conditions of contaminated sites [14]. To date, soluble chromate reductase resulting from *Pseudomonas putida*, *Escherichia coli*, *Shewanella* sp., *Gluconacetobacter* sp., and *Stenotrophomonas maltophilia* have been identified and their basic characteristics enumerated [14].

Although the basic capabilities of CRB and related enzymes in removing Cr(VI) have been examined, little is known about their differences in engineering applications, such as operating conditions, operating guidelines, applicable targets, and cost analysis. In this study, immobilized *Ochrobactrum* sp. CUST210-1 and immobilized enzyme biosystems were applied to remove Cr(VI) from actual wastewater to compare their operational characteristics in a continuous-flow mode. The results can provide insight into strategies for removing Cr(VI) from various environments.

## 2. Materials and Methods

### 2.1. Materials

The *Ochrobactrum* sp. CUST210-1 was isolated from soil in the vicinity of an electroplating factory (Changhua County, Taiwan) using the spread plate method and then cultivated in Luria–Bertani (LB) broth supplemented with $Na_2Cr_2O_7$ (Li-Hsin chemical company, Hsinchu City, Taiwan) ((final concentration: 300 mg $L^{-1}$ Cr(VI)) in 300-mL Erlenmeyer flasks on a rotary shaker (Sunway scientific company, Taipei City, Taiwan) (250 rpm) at 35 °C under aerobic conditions. To identify the Cr(VI)-reducing bacterium, the cells were lysed, and the DNA was extracted. The 16S rRNA gene sequence of the isolate was compared with the NCBI database using BLASTN, and the closest match to the bacterial isolate was retrieved.

To evaluate the toxicity of Cr(VI)-containing wastewater before and after treatment, a Microtox toxicity assay system (Model 500; Azur Environmental, Newark, DE, USA) comprising *Vibrio fischeri* and a fish poison test using *Pseudorasbora parva* obtained from a lake in New Taipei City, Taiwan were used [17,18]. The synthetic wastewater used for toxicity evaluation contained 1/1000 LB supplemented with a final Cr(VI) concentration of 300 mg $L^{-1}$. All analytic chemicals used in the experiment were of an analytical grade.

### 2.2. Distribution of Chromium (VI) Reduction Products

The *Ochrobactrum* sp. CUST210-1 was cultivated in LB supplemented with 300-mg $L^{-1}$ Cr(VI) at 35 °C under aerobic conditions. The shaking velocity and the pH of the culture were set at 150 rpm and 7.0, respectively. The pH of the culture was adjusted by the addition of 0.1-N HCl or NaOH to the culture. After 24 h of cultivation, 200 mL of culture solution containing the CUST210-1 strain was centrifuged at 6000 rpm for 10 min, and the bacterial pellets were obtained. The sequential extraction procedure was then used to analyze the distribution of Cr(VI) reduction products [19,20]. The bacterial pellets were sequentially extracted with 1-M $MgCl_2$; 0.5-M KF; 1-M NaAc; and a mixture of 0.02-M $HNO_3$, 30% $H_2O_2$, and 3.2-M $NH_4Ac$ to fractionate the Cr(III) compounds into exchangeable/loosely, adsorbed, carbonate, and organically bound forms, respectively. Finally, the residue was acidified with 16-M $HNO_3$ until the dark color disappeared. After centrifugation, $Cr(OH)_3$ was obtained. To assess the intracellular $Cr(OH)_3$, the bacterial pellets were placed in 20-mM phosphate buffer (PB), vibrated at 4 °C by an ultrasonic processor (UP-100, ChromTech, Apple Valley, MN, USA), and centrifuged at 12,000 rpm for 20 min at 4 °C. The precipitate was acidified with 16-M $HNO_3$ until the dark color disappeared.

### 2.3. Separation and Purification of Chromate Reductase

After incubating the CUST210-1 strain at 35 °C for 24 h, the cells were collected by centrifugation at 8000 rpm for 15 min. The bacterial pellets were washed with 20-mM PB, and the suspension was placed in PB solution for vibration at 4 °C by the ultrasonic processor (20 kHz, 130 watt). Then, the suspension was centrifuged at 12,000 rpm for 20 min at 4 °C, and the supernatant was collected for further enzyme purification. To purify the chromate reductase, the supernatant was mixed with 20-mM PB; the active components were separated using a HiPrep Q XL 16/10 column (16 mm × 100 mm, 20-mM PB, 2 mL $min^{-1}$) (GE Healthcare, Piscataway, NJ, USA), and raw chromate reductase was collected using a HiTrap Phenyl Sepharose column (7 mm × 25 mm, 50-mM sodium phosphate and 1.0-M ammonium sulphate, 1 mL $min^{-1}$) (GE Healthcare) and then purified using a Mono Q HR 5/5 column (5 mm × 50 mm, 20-mM Bis-Tris, 0.5 mL $min^{-1}$) (GE Healthcare).

To analyze the molecular weight of the chromate reductase, SDS-PAGE was performed on 10% gels, as described by Laemmli [21]. To determine the crystal structure of the chromate reductase, the purified protein was dissolved in Tris-HCl and NaCl (10 and 150 mM, respectively, pH 8.0) and crystallized using the sitting drop vapor diffusion method, with a PEG8000 (10%), NaCl (50 mM), and calcium acetate (100 mM) solution as the precipitant. X-ray diffraction data of the crystals were

collected at a resolution of 2.2 Å (Instrumentation Center, National Taiwan University, Taiwan), and the structure was solved through the molecular replacement method [22]. Homology searches were performed using BLASTP.

## 2.4. Analysis of Genes Involved in the Cr(VI) Reduction

To identify the functional genes involved in the Cr(VI) reduction, known chromate reductase genes were used as a reference. The genes included *apcA*, *nfsA*, *nfsB*, *yieF*, *azr*, *crS*, *mreG*, *hydC*, *mtrC*, *omcA*, and *chrR* [8]. Their primer pairs were used (Table 1), and gene amplification was performed to synthesize corresponding chromate reductase genes from the strain by using a Bio Rad C1000 thermocycler (Bio Rad, Hercules, CA, USA), as previously described [22–30]. The operating conditions for polymerase chain reactions and 454 pyrosequencing followed those of De Silva et al. [31]. The synthesized genes from the CUST210-1 strain were sequenced and their sequences compared with other chromate reductase genes in GenBank.

**Table 1.** Oligonucleotide primer pairs used to amplify the chromate reductase gene.

| Targeted Gene | Sequence Primers (5′–3′) |
| --- | --- |
| *apcA* | F ATGAGTATCGTCACTAAATCCATCG<br>R TACTGCATTGCACCGACAAC |
| *nfsA* | F ATCGAATTCAGACTGAAGGCTCACTTTGC<br>R ATCGCGGATCCACGTAACGCTTTGTCGGT |
| *nfsB* | F GTAGGATCCGATATCATTTCTGTCGC<br>R ACTGAATTCTTACACTTCGGTTAAGGTG |
| *yieF* | F AGCTCATTTAATGGCATGG<br>R ATCAAGGGAATGTCGGCAA |
| *azr* | F AATACGGTAAGCGCAGCG<br>R ATTATGTAAACCTATTTG |
| *crS* | F CATATGGCCTTGCTCTTCACCCCCCTGGAACTC<br>R GAATTCCTAAAACCCCCT<br>TTGGTACTGGGGGGGTAC |
| *mreG* | F ATCACTTCGGAACTGGGTGT<br>R TACCCCGCAACACACTGTAA |
| *hydC* | F CCTCTTTATCTTTAACAAAGGGTGCAGGGE<br>R GGGTGCAGGGTTCAGCGAGCCTCTTTTTGGG |
| *mtrC* | F AGATCTGTTGGCGCTAGAGCATAG<br>R GCGGCCGCTAATAGGCTTCCCAATTTGT |
| *omcA* | F AGCCGTATGATAGTGGGCTG<br>R TCACTGAGACGAATACGGCG |
| *chrR* | F ATGTCTGATACGTTGAAAGTTGTTA<br>R CAGGCCTTCACCCGCTTA |

## 2.5. Toxicity Evaluation of Cr(VI) Reduction Products in Batch Treatment

To evaluate the change in its toxicity after treatment, 300-mg L$^{-1}$ Cr(VI)-containing wastewater was treated by the CUST210-1 strain or chromate reductase in batch mode. The initial concentrations of the CUST210-1 strain and the enzymatic activity were $3 \times 10^7$ cfu mL$^{-1}$ and 10.8 U mg-protein$^{-1}$, respectively. The treatment time for the CUST210-1 strain and chromate reductase was set for 2 h and 0.5 h, respectively. Toxicity was evaluated using *V. fischeri* and *P. parva*, and toxicity units are expressed as EC$_{50}$ and LC$_{50}$. The analysis data were evaluated using a Probit analysis in IBM SPSS, version 20 (IBM Corp, Armonk, NY, USA).

### 2.6. Apparatus for Continuous Cr(VI) Removal

A cylindrical packed-bed bioreactor (CPB; length 25 cm, id 16 cm) was constructed from acrylic materials for the immobilized *Ochrobactrum* sp. CUST210-1 biosystem (Figure 1A). Three sampling pores were constructed at the top of the biosystem through which the pH value, dissolved oxygen (DO) content, and Cr(VI) concentration in the wastewater could be measured. Plastic Raschig rings (rosette type, id 2 cm) were used as packing material. The inflow solution, which contained 1/1000 LB supplemented with 300 mg $L^{-1}$ Cr(VI) and the CUST210-1 strain ($3 \times 10^7$ cfu $mL^{-1}$) stored in the reservoir, was continuously recirculated with the flow directed upward from the inlet at the bottom of the reactor. The liquid retention time (LRT) of the inflow solution was controlled at 12 h. When the outlet Cr(VI) concentration was less than 0.5 mg $L^{-1}$, the cell immobilization procedure was regarded as completed.

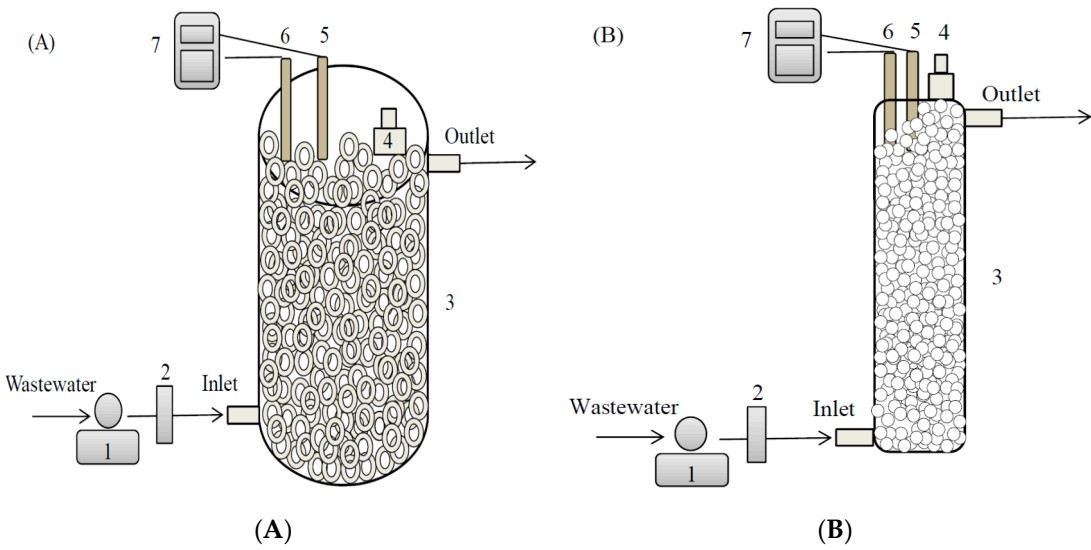

**Figure 1.** Schematic of (**A**) immobilized *Ochrobactrum* sp. CUST210-1 biosystem and (**B**) immobilized enzyme biosystem: (1) peristaltic pump, (2) flow meter, (3) bioreactor, (4) sampling port, (5) pH electrode, (6) Dissolved oxygen (DO) electrode, and (7) digital pH/DO meter.

Calcium alginate beads are made through external gel formation [32]. Three percent (*w/v*) sodium alginate in 50-mM sodium PB (pH 7.0) was mixed with 5% chromate reductase solution at the ratio of 1:1. The mixture was taken into a syringe and dripped directly into a gently stirred 1.5% $CaCl_2$ solution from a height of approximately 2 cm; this resulted in the formation of spherical beads of enzyme-entrapped calcium alginate. The beads were left in the $CaCl_2$ solution for 30 min, recovered by filtration, and thoroughly washed with distilled water thrice. Toxicity evaluation of the Cr(VI) reduction products was conducted using the immobilized chromate reductase beads in batch mode. The beads were packed into a cylindrical column bioreactor (CCB) until the immobilized enzyme biosystem was used for continuous Cr(VI) removal. The CCB (length 32 cm, id 8 cm) was constructed from acrylic materials, with sampling pores installed as in the *Ochrobactrum* sp. CUST210-1 biosystem (Figure 1B).

### 2.7. Effects of Operating Parameters on Continuous Cr(VI) Removal by Immobilized Ochrobactrum sp. CUST210-1 Biosystem

When the *Ochrobactrum* sp. CUST210-1 biosystem completed immobilization, synthetic wastewater was continuously introduced to the CPB. The effects of different operating parameters, including LRT (1–6 h), pH (7–9), operating temperature (25 °C–45 °C), and Cr(VI) concentration (50–400 mg $L^{-1}$), on continuous Cr(VI) removal were assessed, and the synthetic wastewater (1/1000 LB supplemented

with Cr(VI)) was used in this study. The designed LB concentration simulated organic concentrations in tannery wastewater after treatment by chemical precipitation. Generally, the inflow Cr(VI) concentration, pH, LRT, and operating temperature were set as 200 mg L$^{-1}$, 7.0, 4 h, and 35 °C, respectively. Each operating parameter was continuously operated for at least 6 h, with the results expressed as the average for each parameter.

Furthermore, the effects of the actual Cr(VI)-containing wastewater, which was collected from the effluent of tannery wastewater after treatment by chemical precipitation (New Taipei City, Taiwan), on Cr(VI) removal were evaluated by continuously feeding the water into the biosystem for 196 days. In the case of this actual wastewater, the LRT, operating temperature, and pH were controlled at 4 h, 35 °C, and 7.0.

### 2.8. Effects of Operating Parameters on Continuous Cr(VI) Removal by Immobilized Enzyme Biosystem

The effects of various operating parameters, including the LRT (0.5–3 h), operating temperature (25 °C–45 °C), Cr(VI) concentration (25–150 mg L$^{-1}$), and life cycle of the immobilized bead, on continuous Cr(VI) removal by the immobilized enzyme biosystem were assessed with regard to the synthetic wastewater (1/10,000 LB supplemented with Cr(VI)). The designed LB concentration simulated the organic concentration in electroplating wastewater after treatment by chemical precipitation. The initial enzyme activity, inflow Cr(VI) concentration, LRT, pH, and operating temperature were 10.2 U mg-protein$^{-1}$, 100 mg L$^{-1}$, 1 h, 7, and 35 °C, respectively, unless otherwise stated. Each parameter was continuously operated for at least 6 h, with results expressed as the average for each. Furthermore, the effects of actual Cr(VI)-containing wastewater, derived from the effluent of electroplating wastewater after treatment by chemical precipitation (Changhua County, Taiwan), on continuous Cr(VI) removal were evaluated by feeding the water into the biosystem for 196 days. In this case, the LRT, operating temperature, and pH were controlled at 1 h, 35 °C, and 7.0, respectively. The influent pH was adjusted to the desired working pH level by the addition of 1-N HCl or NaOH.

### 2.9. Analysis

Na$_2$Cr$_2$O$_7$ of special-grade chemicals was dried at 200 °C for 1 h and left in a desiccator. Subsequently, 100.8 mg of Na$_2$Cr$_2$O$_7$ was weighed, dissolved in water, and diluted to 100 mL. The diluted solution was used as a standard stock solution of 400-mg L$^{-1}$ Cr(VI). The colorimetric method (1,5-diphenylcarbazide method) for Cr(VI) measurement was used as described previously [11]. The concentrations of collected extracts for various forms of Cr species and total Cr were analyzed using an atomic absorption spectrophotometer (Hitachi, Tokyo, Japan). DO and pH values were continuously determined by an online DO and pH meter (Hanna Instruments, Woonsocket, RI, USA). For Biochemical oxygen demand (BOD) analysis, the standard method 5210B was adopted. Chemical oxygen demand (COD) was analyzed using a Hach DR2800 portable spectrophotometer (Hach, Loveland, CO, USA). For toxicity evaluation, the inhibition reaction of bioluminescence of *V. fischeri* exposed to tested solutions for 15 min was analyzed, and the acute toxic reaction of *P. parva* exposed to the solutions for 96 h was analyzed.

## 3. Results and Discussion

### 3.1. Distribution of Cr(VI) Reduction Products and Possible Cr(VI) Reduction Mechanism

Knowledge of the state, form, and location of Cr(VI) treated by *Ochrobactrum* sp. CUST210-1, such as adsorption state, complex formation, or precipitate form, is essential for the development of a suitable remediation process. In addition, the characterization of the product is helpful for understanding the reduction mechanism of Cr(VI) by *Ochrobactrum* sp. CUST210-1. Table 2 lists the distribution of various chromium species inside and outside the cells after the batch treatment. A small amount (3.05%) of CrO$_4$$^{2-}$ or Cr$_2$O$_7$$^{2-}$ was adsorbed on the positively charged functional group of the cell wall, 8.49% as the Cr(III) form was coordinated with negatively charged functional groups such as carbonyl or amide

groups, and 3.12% as the Cr(III) form existed in the solution as a free form; however, most of the Cr(VI) was deposited, inside or outside of the cells, as the $Cr(OH)_3$ form during the Cr(VI) reduction due to Cr(III) tending to form precipitates at pH > 5 [4]. Table 3 lists the Cr(III) distribution in Cr(III) species of Cr(VI) reduction products after the batch treatment. The results indicate that the exchangeable form of Cr(III), adsorbed Cr(III), organically bound Cr(III), and carbonate-bound Cr(III) on the cell surface accounted for 0.21%, 2.90%, 5.53%, and 0.13%, respectively. $Cr(OH)_3$ precipitate accounted for 88.01%. In brief, Cr(VI) compounds were usually adsorbed by the cell surface, and Cr(III) may exist in the form of organic-Cr(III) complexes or in the precipitate form of $Cr(OH)_{3(s)}$. We speculate that few complex salts occurred in the form of $Cr(OH)_4^-$, $Cr_2O_2(OH)_4^{2-}$, or $Cr_3O_4(OH)_4^{3-}$ in the solution. Moreover, most $Cr(OH)_{3(s)}$ was located inside cells, which was similar to results reported for Cr(VI) reduction by *Bacillus amyloliquefaciens*, *Synechocystis* sp., *Cellulosimicrobium funkei*, and *P. aeruginosa* [33–36] but different from studies identifying extracellular deposits by *Ochrobactrum anthropi*, *Vigribacillus* sp., and *Shewanella oneidensis* [37–39].

**Table 2.** Distribution of chromium (VI) reduction products.

| Cr Distribution | Cr(VI) by Surface Adsorption | Cr(III) by Surface Adsorption | Cr(III) in Solution | Cr(VI) in Solution | $Cr(OH)_{3(s)}$ Outside Cell | Cr(VI) Inside Cell | $Cr(OH)_{3(s)}$ Inside Cell |
|---|---|---|---|---|---|---|---|
| Relative amount (%) | 3.05% | 8.49% | 3.12% | 0% | 26.70% | 0% | 58.64% |

**Table 3.** Distribution of Cr(III) species in Cr(VI) reduction products.

| | Cr(III) in Solution | Cr(III) Exchangeable Form | Cr(III) Adsorbed Form | Cr(III) Organically Bound | Cr(III) Carbonate Form | $Cr(OH)_{3(s)}$ Outside Cell | $Cr(OH)_{3(s)}$ Inside Cell |
|---|---|---|---|---|---|---|---|
| Relative amount of Cr(III) species (%) | 3.22% | 0.21% | 2.90% | 5.53% | 0.13% | 27.53% | 60.48% |

On the basis of the product analysis results, the following Cr(VI) reduction mechanism or Cr(VI) resistance-tolerance mechanism of *Ochrobactrum* sp. CUST210-1 may be inferred. (1) Partial Cr(VI) directly binds via electrostatic attraction to positively charged functional groups on the surface of the CUST210-1 strain (3.05%), or partial Cr(VI) is reduced to Cr(III) via membrane-associated chromate reductase or soluble chromate reductase (i.e., ChrR) and binds to negatively charged functional groups on the surface of the CUST210-1 strain (8.49%), (2) a portion of Cr(VI) is reduced to Cr(III) by ChrR and (i) remains in the solution as the $Cr(OH)_4^-$ form or even as the multinuclear polymers $Cr_2O_2(OH)_4^{2-}$ or $Cr_3O_4(OH)_4^{3-}$ (3.12%) or (ii) is mainly formed as $Cr(OH)_{3(s)}$ outside the cell (26.70%), and (3) most Cr(VI) species enter the cell via the $SO_4^{2-}$ channels then are reduced to Cr(III) via intracellular ChrR and deposited inside the cell as $Cr(OH)_{3(s)}$ (58.64%). An analogous mechanism has been discovered in other bacteria, such as P. aeruginosa and S. maltophilia [14,36].

*3.2. Enzyme and Genes Involved in the Cr(VI) Reduction*

According to the literature, chromate reductase is divided into the following categories: (1) Cytochrome C: cyt $c_7$, mtrC, and omcA have been identified in *Desulfuromonas* and *Shewanella*. (2) Flavin protein: ChrR, nfsA, and yieF have been found in *E. coli*, *Pseudomonas*, *Gluconacetobacter*, *Shewanella*, and *Bacillus*. (3) Old yellow enzymes: CrS has been discovered in *Thermus scotoductus*. (4) Hydrogenases: mreG and hydC have been found in *Desulfovibrio* [7]. The chromate reductase of *Ochrobactrum* sp. CUST210-1 was found at approximately 25 kDa on the SDS-PAGE gel, suggesting that the chromate reductase was the ChrR protein; similar results were reported for *S. maltophilia* and *Alishewanella* sp. induced in the presence of Cr(VI) [14,40]. To confirm this speculation, the 11 primer pairs of known chromate reductase genes were used to synthesize corresponding genes involved in Cr(VI) reduction by the CUST210-1 strain [8]. Results indicate that the *chrR* gene was successfully synthesized. The BLASTN analysis of the sequence indicated 99.2%, 97.2%, 96.8%, 94.8%,

and 92.6% similarity with the ChrR sequences from *E. coli*, *P. putida*, *Shewanella* sp., *Gluconacetobacter* sp., and *S. maltophilia*, respectively.

Theoretical ChrR protein sequences derived from the nucleotide sequence revealed an identical amino acid sequence for *Ochrobactrum* sp. CUST210-1. Figure 2 displays the crystal structure of ChrR in *Ochrobactrum* sp. CUST210-1. The ChrR includes three β-sheets, 14 α-helices, and five loops; BLASTP analysis revealed 98.5%, 96.8%, 95.2%, 91.5%, and 86.8% homology to the ChrR of *E. coli*, *P. putida*, *Shewanella* sp., *Gluconacetobacter* sp., and *S. maltophilia.*

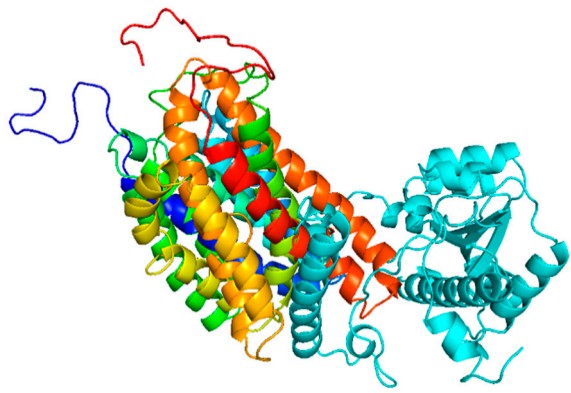

**Figure 2.** Crystal structure of ChrR in *Ochrobactrum* sp. CUST210-1.

### 3.3. Toxicity Evaluation of Cr(VI) Reduction Products in Batch Treatment

When 300-mg $L^{-1}$ Cr(VI)-containing wastewater was treated by *Ochrobactrum* sp. CUST210-1 and ChrR in batch, Cr(VI) removal efficiencies achieved 62.5% and 48.5%, respectively. The residual Cr(VI) concentrations were 112.5 mg $L^{-1}$ with the CUST210-1 treatment and 154.2 mg $L^{-1}$ with the ChrR treatment, respectively. Before treatment, Microtox toxicity ($EC_{50}$) and P. parva ($LC_{50}$) were 18.2–18.5 mg $L^{-1}$ and 40.5–40.8 mg $L^{-1}$, respectively. After treatment by the CUST210-1 strain, the toxicities of Microtox and P. parva were decreased to 62.4 mg $L^{-1}$ and 95.8 mg $L^{-1}$. After treatment by ChrR, the toxicities of Microtox and P. parva were decreased to 86.2 mg $L^{-1}$ and 123.5 mg $L^{-1}$, respectively. For *Ochrobactrum* sp. CUST210-1 and ChrR, respectively, 57.8–70.3% and 67.0–78.9% detoxification efficiencies were achieved. Thus, the ChrR had higher detoxification efficiency than the bacterial cells. Notably, although the removal efficiency of the CUST210-1 strain was higher than that of ChrR protein, the reverse was true for detoxification efficiency. This may be attributed to the Cr(VI) product from the ChrR treatment being relatively simpler than that of the bacteria [5]. It also indicates the advantage and potential of the Cr(VI) treatment by ChrR.

### 3.4. Effects of Operating Parameters on Continuous Cr(VI) Removal by Immobilized Ochrobactrum sp. CUST210-1 Biosystem

Figure 3A presents the effect of LRT on the continuous treatment of synthetic Cr(VI)-containing wastewater by the immobilized *Ochrobactrum* sp. CUST210-1 biosystem. The results indicate that the removal efficiency for Cr(VI) gradually increased, but the removal capacity for Cr(VI) gradually decreased as the LRT increased. When the LRT was 4 h, the removal efficiency achieved 100%, and the removal capacity achieved an acceptable level of $1.71 \pm 0.2 \times 10^{-9}$ mg-Cr(VI) $(\text{cfu-h})^{-1}$. Thus, the operating LRT of the experiment was set at 4 h. Temperature and pH play a key role in the growth of bacterial strains; hence, their effects on the removal characteristics for Cr(VI) were evaluated. Figure 3B represents the effect of pH on the continuous treatment of synthetic Cr(VI)-containing wastewater. A pH of 7–8.5 had a nonsignificant effect ($p > 0.05$) on the removal efficiency for Cr(VI), and the removal efficiency achieved >99.2% ± 0.1%. Moreover, a pH of 7–9 had a nonsignificant effect ($p > 0.05$) on the removal capacity for Cr(VI), and the removal capacity was maintained in the range of $1.60 \pm 0.05$ to $1.67 \pm 0.01 \times 10^{-9}$ mg-Cr(VI) $(\text{cfu-h})^{-1}$. To consider the characteristics of

Cr(VI)-containing wastewater and the operational performance of the biosystem, the pH value of the experiment was controlled at 7.0. An optimal Cr(VI) reduction at pH 7.0 was determined for Bacillus sp., P. aeruginosa, and S. maltophilia [14,36,41]. The decrease in Cr(VI) reduction efficiency at pH 9.0 was mainly attributed to the possibility that deviation from appropriate pH values may affect the degree of ionization of ChrR [36]. These results indicate the *Ochrobactrum* sp. CUST210-1 biosystem had good adaptability to pH changes. Figure 3C displays the effect of the operating temperature on the continuous treatment of synthetic Cr(VI)-containing wastewater. An operating temperature between 30 °C and 40 °C had a nonsignificant effect ($p > 0.05$) on the removal efficiency, and an efficiency >99.5% was achieved. By contrast, Cr(VI) reduction by Serratia rubidaea and C. funkei was greatly influenced by temperature in a similar temperature range [35,42]. These results indicate the *Ochrobactrum* sp. CUST210-1 had a high adaptability to temperature changes compared with other bacterial strains. Although the removal efficiency of the biosystem achieved 99.6% ± 0.08% at 40 °C, the outlet Cr(VI) concentration in the effluent was 0.8 ± 0.16 mg $L^{-1}$—meeting the current industrial discharge standards for Cr(VI) was difficult. With a further increase of the operating temperature, Cr(VI) removal was decreased to 96.2% ± 0.15%, presumably due to the loss of viability or metabolic activity of the CUST210-1 strain. Thus, the operating temperature of the following experiment was controlled at 35 °C. Figure 3D represents the effect of the inlet Cr(VI) concentration on the continuous treatment of synthetic Cr(VI)-containing wastewater. The removal capacity increased linearly with the inlet Cr(VI) concentration, indicating that the system processed a higher removal capacity, with no chromium toxicity to the CUST210-1 biosystem [36]. Moreover, the removal efficiency maintained a high efficiency until 350 mg $L^{-1}$. *Vigribacillus* sp. was reported to completely reduce Cr(VI) at 150 mg $L^{-1}$ after 100 h in a batch operation [38]. Acinetobacter junii reduced 54-mg $L^{-1}$ Cr(VI) with 99.95% removal efficiency after 2 h in a batch operation [43]. C. funkei removed 250-mg $L^{-1}$ Cr(VI) with 80.43% removal efficiency after 120 h of exposure in a batch operation [35], whereas S. maltophilia could treat 500-mg $L^{-1}$ Cr(VI) with 92% removal efficiency after 120 h of exposure in a batch operation [14]. By comparison, *Ochrobactrum* sp. CUST210-1 achieved a competitive efficiency in removing Cr(VI) in a continuous operation. Although the efficiency was high at 99.5% ± 0.13% when feeding with 350-mg Cr(VI) $L^{-1}$, the outlet Cr(VI) concentration was 1.75 ± 0.46 mg $L^{-1}$, exceeding the current discharge limit of 0.5 mg $L^{-1}$. To extend the treatment range of the Cr(VI) concentration, more details (e.g., DO control and turbidity pretreatment) for better performance were investigated by feeding with actual wastewater.

### 3.5. Effects of Operating Parameters on Continuous Cr(VI) Removal by Immobilized ChrR Biosystem

Figure 4A depicts the effect of the LRT on the continuous treatment of synthetic Cr(VI)-containing wastewater by the immobilized enzyme biosystem. The removal efficiency increased with the LRT, and the removal capacity decreased as the LRT increased. Optimal LRT occurred at 1 h—at which point, the removal efficiency and capacity of the biosystem achieved 100% and 100 ± 1.2 mg-Cr(VI) $L^{-1}$ $h^{-1}$, respectively. Thus, the operating LRT of the experiment was 1 h. Figure 4B portrays the effect of the operating temperature on the continuous treatment of synthetic Cr(VI)-containing wastewater. The ChrR biosystem exhibited a high removal efficiency (>98.20% ± 0.2%) over a wide range of temperatures (25 °C–40 °C), indicating the immobilized ChrR had high stability in a wide temperature range. Although the efficiency of 99.65% ± 0.07% at 25 °C seemed satisfactory, the outlet Cr(VI) concentration of 0.35 ± 0.07 mg $L^{-1}$ exceeded the maximum contaminant level of 0.05 mg $L^{-1}$ for fisheries, drinking, ground, and surface waters. A higher temperature (e.g., 45 °C) could impair the function of ChrR and dramatically reduce the Cr(VI) removal efficiency [44]. Thus, the operating temperature of the experiment with the immobilized ChrR biosystem was controlled at 35 °C. Figure 4C presents the effect of the inlet Cr(VI) concentration on the continuous treatment of synthetic Cr(VI)-containing wastewater. The removal efficiency was excellent (>99.96%) at 25–150 mg $L^{-1}$. The maximum outlet Cr(VI) concentration was only 0.045 mg $L^{-1}$, even at 150-mg $L^{-1}$ Cr(VI) feeding, which meets industry effluent standards and maximum contaminant levels. Figure 4D displays the

effect of the operating time on the continuous treatment of the synthetic wastewater. The removal efficiency initially maintained a high level but gradually decreased with the increased operating time. After 35 days of operation, the efficiency curve exhibited an obvious turn. On day 42, the outlet Cr(VI) concentration was $3.68 \pm 0.4$ mg L$^{-1}$, exceeding maximum contaminant levels. Thus, regular replacement (or frequency of use) during a 35-day cycle of immobilized ChrR beads should be considered to maintain high removal capabilities in a long-term operation. To our knowledge, this is first report of an immobilized ChrR biosystem applied to continuous Cr(VI) removal.

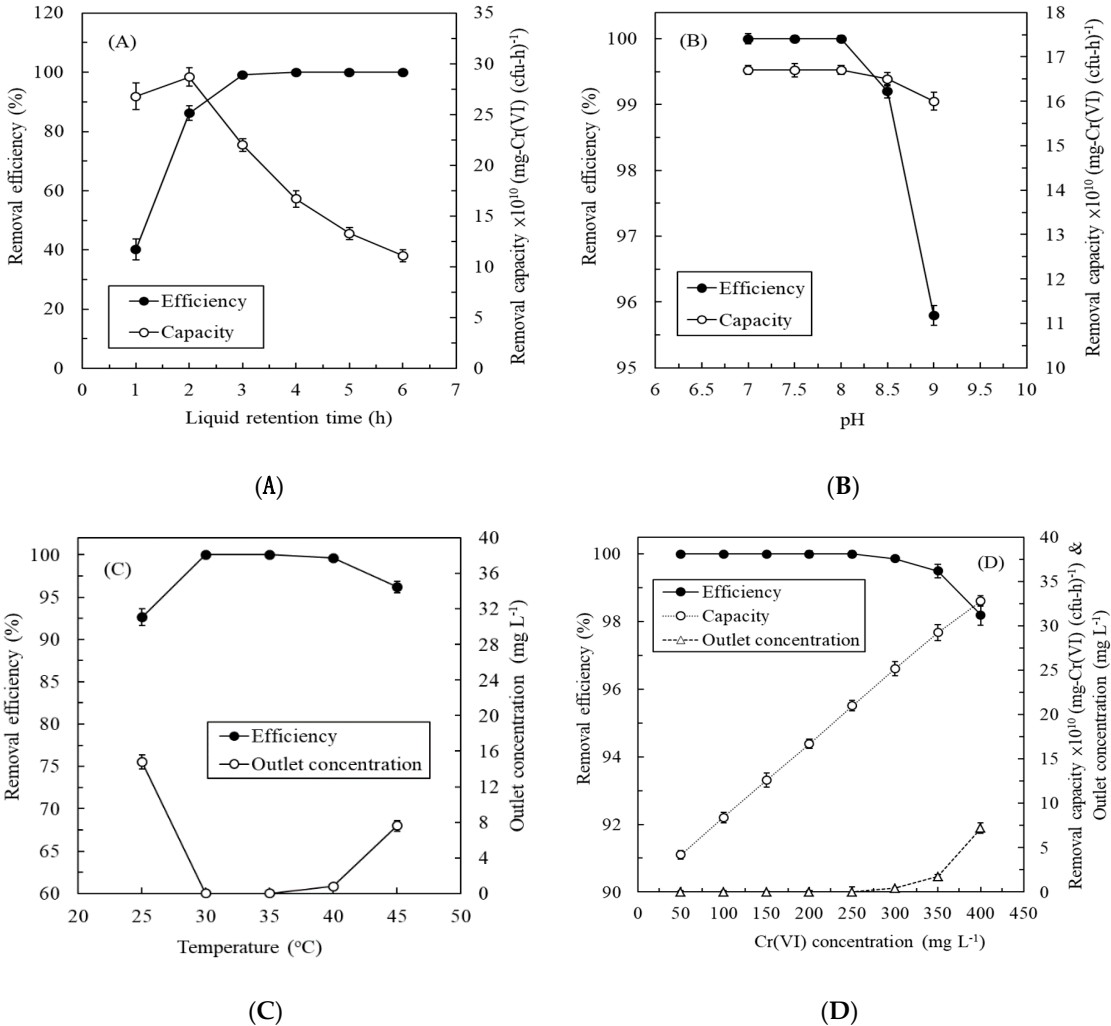

**Figure 3.** Effects of (**A**) the liquid retention time, (**B**) pH, (**C**) temperature, and (**D**) Cr(VI) concentration on the continuous treatment of synthetic Cr(VI)-containing wastewater by the immobilized *Ochrobactrum* sp. CUST210-1 biosystem.

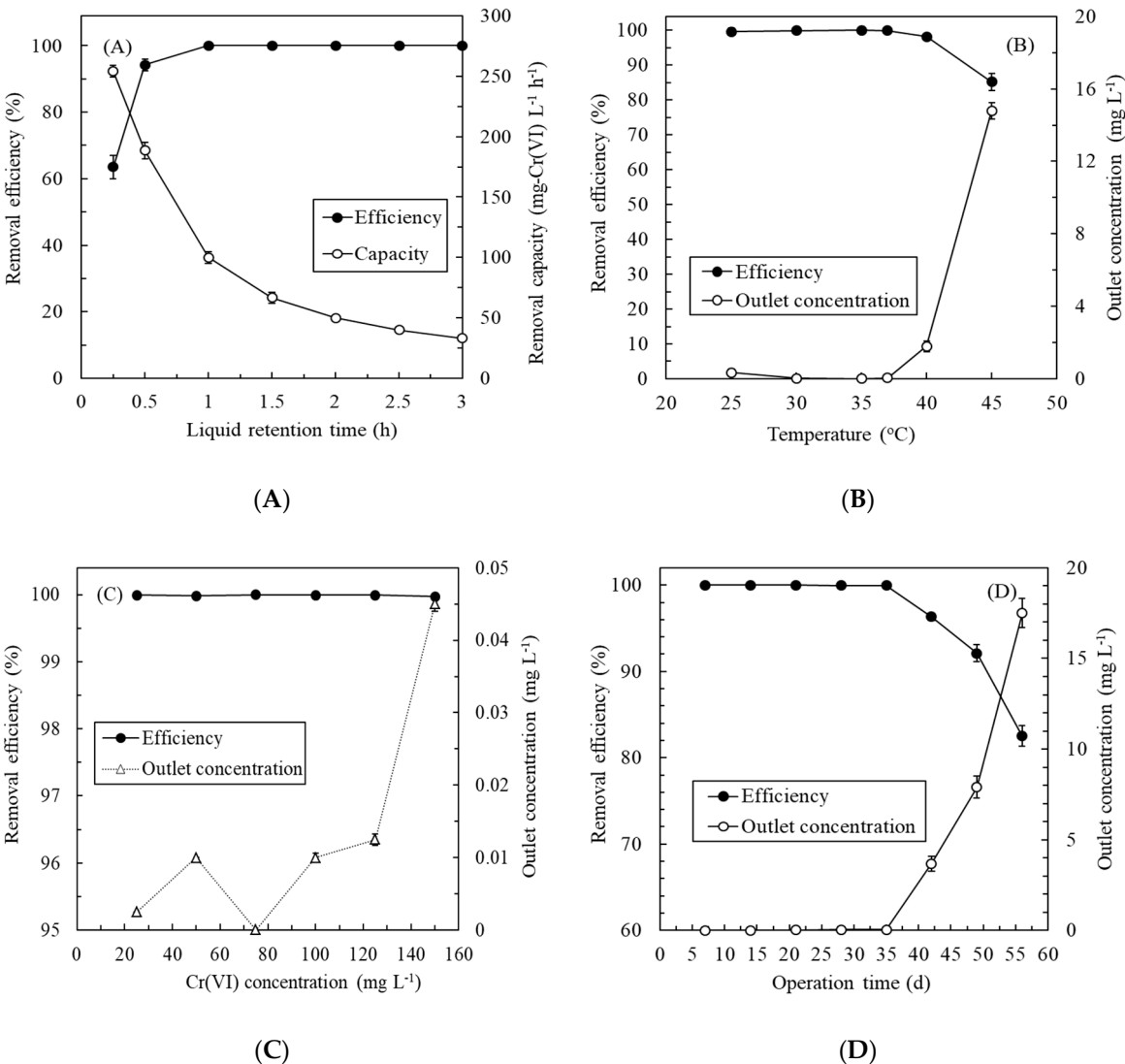

**Figure 4.** Effects of (**A**) the liquid retention time, (**B**) temperature, (**C**) Cr(VI) concentration, and (**D**) operation time on the continuous treatment of synthetic Cr(VI)-containing wastewater by the immobilized ChrR biosystem.

### 3.6. Continuous Treatment of Actual Cr(VI)-Containing Wastewater by Immobilized Ochrobactrum sp. CUST210-1 Biosystem and Immobilized ChrR Biosystem

Actual Cr(VI)-containing wastewater derived from tannery or electroplating wastewater was continuously treated by our biosystems for 196 days. Average $S^{2-}$ concentration, suspended solid (SS), BOD, and COD in the tannery wastewater were $2.5 \pm 0.6$ mg $L^{-1}$, $303 \pm 23$ mg $L^{-1}$, $158 \pm 8$ mg $L^{-1}$, and $261 \pm 15$ mg $L^{-1}$, respectively. According to the previous results, the inlet pH of $9.2 \pm 0.6$ was adjusted to pH $7.0 \pm 0.5$ by a pH adjustment system. Figure 5A depicts the continuous treatment of actual tannery wastewater by the immobilized *Ochrobactrum* sp. CUST210-1 biosystem. The inlet Cr(VI) concentration varied from 284 to 471 mg $L^{-1}$ during the 196-day treatment period. The change of luminescent *V. fischeri* toxicity (luminescence inhibition) in response to the water quality and the dynamic of the outlet Cr(VI) concentration were similar. This suggested that the residual Cr(VI) concentration was associated with the toxicity of the wastewater. The outlet Cr(VI) concentration exceeded the discharge limit of 0.5 mg $L^{-1}$ on day 14, day 56, day 63, day 91, day 154, day 189, and day 196. Although the inlet concentration was only $295 \pm 1.7$ mg $L^{-1}$ (below the ideal treatment concentration, 300 mg $L^{-1}$) on day 91, the removal efficiency dropped to $98.52\% \pm 0.04\%$, with an outlet concentration of $4.37 \pm 0.09$ mg $L^{-1}$ due to turbidity problems. When a filter device was installed, even with an inlet concentration as

high as 425 ± 10 mg L$^{-1}$ (on day 133), the removal efficiency achieved 99.91% ± 0.04%, with an outlet concentration of 0.38 ± 0.17 mg L$^{-1}$. When 408 ± 3.75-mg L$^{-1}$ Cr(VI) was introduced to the biosystem on day 154, the removal efficiency dropped to 97.86% ± 0.27%, with an outlet concentration of 8.73 ± 1.16 mg L$^{-1}$ because of an insufficient DO supply (3.6 ± 0.4 mg L$^{-1}$). Thus, an aeration device and a DO sensor were installed (DO would be controlled at >4.0 mg L$^{-1}$), and the removal efficiency gradually rose and stabilized. With the filter device, aeration device, and DO sensor, the outlet Cr(VI) concentration would meet the industry discharge limit of 0.5 mg L$^{-1}$ if the inlet Cr(VI) concentration was below 430 mg L$^{-1}$.

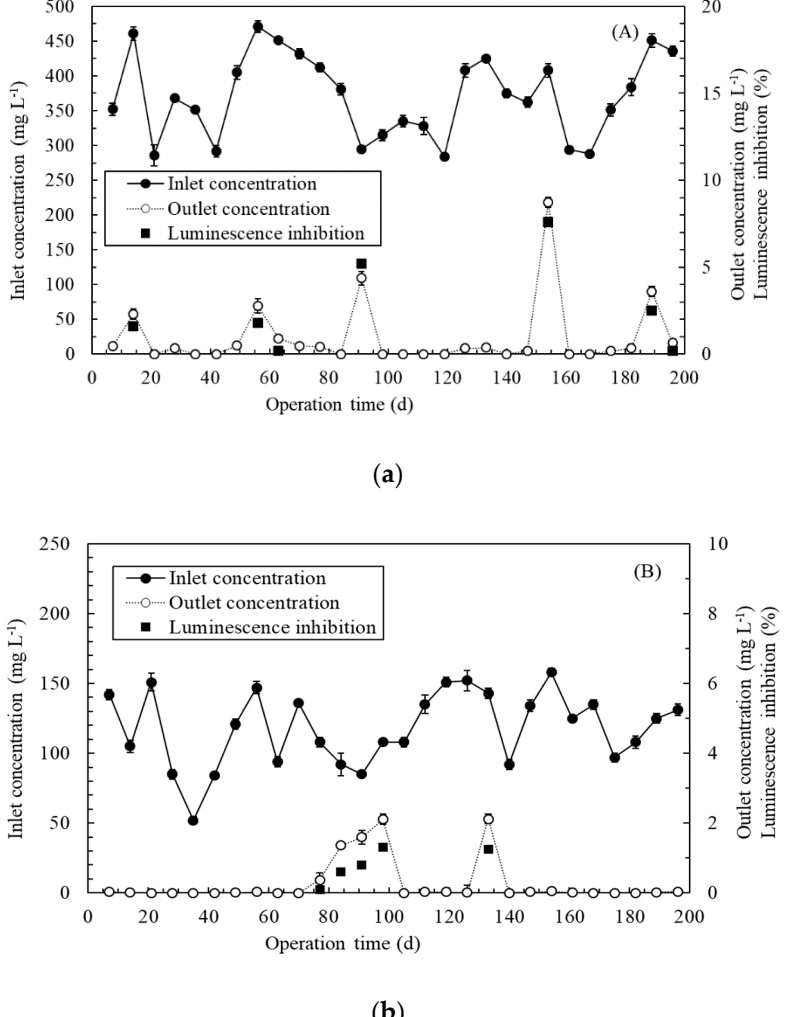

(**a**)

(**b**)

**Figure 5.** Continuous treatment of actual Cr(VI)-containing wastewater by (**a**) the immobilized *Ochrobactrum* sp. CUST210-1 biosystem and (**b**) immobilized ChrR biosystem.

Figure 5B represents the continuous treatment of actual electroplating wastewater by the immobilized ChrR biosystem. The average Ni$^+$ concentration, Na$^+$ concentration, SO$_4^{2-}$ concentration, Cl$^-$ concentration, SS, BOD, and COD in the wastewater were 13.5 ± 0.6 mg L$^{-1}$, 4.2 ± 0.8 mg L$^{-1}$, 6.5 ± 0.7 mg L$^{-1}$, 4.8 ± 0.3 mg L$^{-1}$, 24 ± 2.3 mg L$^{-1}$, 42 ± 5.6 mg L$^{-1}$, and 68 ± 2.7 mg L$^{-1}$, respectively. According to the previous results, a pH adjustment system was installed to adjust the pH of the water from 5.1 ± 0.8 to 7.0 ± 0.5. The inlet Cr(VI) concentration varied from 52 to 158 mg L$^{-1}$ during the 196-day treatment period. The trend between the changes in the luminescent bacteria toxicity for water quality and the dynamics of the outlet Cr(VI) concentration was consistent. To evaluate the adaptability of the biosystem, the immobilized enzyme replacement was deliberately neglected on the

70th day, because the replacement frequency was set to a 35-day cycle. The outlet Cr(VI) concentration exceeded the maximum contaminant level of 0.05 mg $L^{-1}$ on days 77–98. On day 105, the immobilized ChrR beads were replaced, and the removal efficiency gradually stabilized to >99.97%. Moreover, the outlet concentration was less than 0.05 mg $L^{-1}$. When 143 ± 0.65-mg $L^{-1}$ Cr(VI) was fed on day 133, the removal efficiency dropped to 98.52% ± 0.05%, with an outlet concentration of 2.11 ± 0.08 mg $L^{-1}$ because of an insufficient DO supply (2.8 ± 0.6 mg $L^{-1}$). Thus, an aeration device and a DO sensor were installed, and DO in the wastewater was controlled at >3.0 mg $L^{-1}$. The removal efficiency of the immobilized ChrR biosystem gradually stabilized at 99.98–100%, and the outlet Cr(VI) concentration conformed to the maximum contaminant level of 0.05 mg $L^{-1}$. Although the maximum contaminant standards for waterbody and wastewater discharges differ at present, developing effective technology to meet future, more stringent standards is worthwhile.

In the literature, limited studies have focused on the treatment of actual Cr(VI)-containing wastewater by a continuous biosystem. Rida et al. (2012) utilized an immobilized O. intermedium batch system to remove 268-mg $L^{-1}$ Cr(VI) from artificial and industrial sewage waters and achieved 65% and 91.2% removal efficiencies, respectively, within 72 h [45]. Chen et al. (2016) applied a microbial fuel cell to remove 300-mg $L^{-1}$ Cr(VI) from synthetic wastewater under anaerobic conditions and achieved a 96.5% removal efficiency [11]. Jin et al. (2017) utilized a sequencing batch reactor to remove 80-mg $L^{-1}$ Cr(VI) from synthetic wastewater and achieved an 81.3% removal efficiency [4]. Our immobilized *Ochrobactrum* sp. CUST210-1 and immobilized ChrR biosystems substantially reduced the Cr(VI) concentrations of <425 mg $L^{-1}$ and <150 mg $L^{-1}$, respectively, in actual Cr(VI)-containing wastewater, suggesting that these developed biosystems can be used in the bioremediation of various Cr(VI)-contaminated wastewaters.

*3.7. Characteristics and Economic Analysis of the Biosystems*

In designing a feasible biosystem, the main goal is to meet a performance level and minimize capital and operating costs. Table 4 offers a comparison of economic and operational characteristics of the biological system and the enzyme system. The basis of the evaluation was to scale up the infrastructure of the reactor system by 10 for field applications. The treatment cost was estimated per one-year cost, and related infrastructures, including the reactor itself, piping, filter device, and aeration device, were calculated based on 10-year life amortizations. The electricity cost was calculated at $25 per month. The results indicate the total costs of the immobilized *Ochrobactrum* sp. CUST210-1 biosystem ($6750 $y^{-1}$) were higher than those of the immobilized ChrR biosystem ($6050 $y^{-1}$). This is due in part to the CUST210-1 biosystem's requirement of a filter device to remove suspended solids from tannery wastewater. The biological treatment system can be applied to treat wastewater with high concentrations of Cr(VI) (~425 mg $L^{-1}$), and the effluent can comply with industrial wastewater discharge standards (<0.5 mg $L^{-1}$). The biological treatment system required a long LRT, and thus, the allowable inflow rate was low (Table 5). In addition, a filter device and DO monitoring were required to improve the removal performance during the operating period. By contrast, the immobilized enzyme system was suitable for treating diluted Cr(VI)-containing wastewater (~150 mg $L^{-1}$), and the effluent could comply with strict water quality standards and meet the standards for fisheries and drinking water (<0.05 mg $L^{-1}$). The ChrR biosystem also operated at a short LRT, and thus, the allowable inflow rate was high. However, DO monitoring and regular replacement of ChrR beads were required to improve performance during the operating period. To date, this is the first report to compare the economic and operational characteristics of biosystems for Cr(VI) removal.

**Table 4.** Comparison of economic and operational characteristics between the immobilized *Ochrobactrum* sp. CUST210-1 biosystem and immobilized ChrR biosystem. BOD: Biochemical oxygen demand.

| | Bacterial Culture (USD/y) | Enzyme Extraction & Immobilization (USD/y) | Reactor Installation (USD/y) | Operational Costs [1] (USD/y) | Application Scope |
|---|---|---|---|---|---|
| CUST210-1 | 350 | - | 6000 [2] | 400 | (1) BOD > 150 mg L$^{-1}$ (2) Inlet Cr(VI) conc: 250–425 mg L$^{-1}$ (3) Outlet Cr(VI) conc: <0.5 mg L$^{-1}$ |
| ChrR | 350 | 350 [3] | 5000 | 350 | (1) BOD <50 mg L$^{-1}$ (2) Inlet Cr(VI) conc: 50–150 mg L$^{-1}$ (3) Outlet Cr(VI) conc: <0.05 mg L$^{-1}$ |

[1] Operating costs include electricity consumption, pH adjustment, filter material, and other miscellaneous costs. [2] Includes filter device. [3] Bead replacement (10 times y$^{-1}$).

**Table 5.** Comparison of economic and operational characteristics between the immobilized *Ochrobactrum* sp. CUST210-1 biosystem and immobilized ChrR biosystem. DO: dissolved oxygen.

| | Manpower Requirement & Ease of Operation | Frequency of Sludge Treatment (time y$^{-1}$) | Characteristics |
|---|---|---|---|
| CUST210-1 | Relatively low | 1 | (1) Suitable for treating high Cr(VI) conc. and complying with wastewater discharge standard (2) Suitable for low influent flow rate (3) Requires filter device and DO monitor |
| ChrR | Relatively high | 10 | (1) Suitable for treating low Cr(VI) conc. and complying with stricter environmental standards (2) Allows high influent flow rate (3) Requires DO monitor and regular replacement of enzyme beads |

## 4. Conclusions

The results clearly demonstrate the application potential of an immobilized *Ochrobactrum* sp. CUST210-1 biosystem and an immobilized ChrR biosystem. The *Ochrobactrum* sp. CUST210-1 biosystem demonstrated good adaptability to pH variation and was able to operate at a long LRT with relatively concentrated Cr(VI), but it had a relatively high cost. The *chrR* gene was identified in *Ochrobactrum* sp. CUST210-1; thus, the ChrR protein is responsible for Cr(VI) reduction and results in large amounts of Cr(OH)$_{3(s)}$. The ChrR biosystem exhibited good adaptability to the operating temperature and was able to operate at a short LRT with relatively diluted concentrations, and it was relatively cheap. The *Ochrobactrum* sp. CUST210-1 biosystem is recommended for tannery wastewater treatment, especially for a posttreatment system. The immobilized ChrR biosystem is recommended for electroplating wastewater treatment as a posttreatment system or in preparation for meeting stricter water quality standards in the future. Through economic analysis and a comparison of operation characteristics, we provided insight into strategies for removing Cr(VI) from various environments.

**Author Contributions:** All authors collaborated to carry out the work presented here. Y.-C.C. and G.-H.W. conceived and designed the experiments; T.-H.T., C.-H.C., and C.-Y.C. performed the experiments; Y.-C.C. wrote the paper; and T.-H.T., G.-H.W., and Y.-C.C. reviewed and edited the manuscript. All authors have read and agreed to the published version of the manuscript.

**Funding:** This research was supported by the Ministry of Science and Technology of Taiwan, grant numbers MOST 107-2313-B-157-001-MY2 and MOST 109-2313-B-157-001.

**Acknowledgments:** We thank Chen, P.Y., Chou, K.W., Hsu, H.J., and Chen, C.Y. for helping with partial paperwork and data processing. This manuscript was edited by Wallace Academic Editing.

**Conflicts of Interest:** The authors declare no conflict of interest.

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
