# Peer review of "Operational Characteristics of Immobilized Ochrobactrum sp. CUST210-1 Biosystem and Immobilized Chromate Reductase Biosystem in Continuously Treating Actual Chromium-Containing Wastewater"

_applsci, doi:10.3390/app10175934_

Round 1
Reviewer 1 Report
Summary:
This manuscript, by Wang et al., surveys the economic and operational characteristics of bacteria and related enzyme biosystems for Cr(VI) removal. Overall, the topic is of interest to applied sciences (MDPI) readers, and the methods are sound. The manuscript is well written and the experiments are properly executed. I suggest some minor revisions before this manuscript is acceptable for publication.
Broad Comments:
- Although the introduction talks clearly about the need for the study. Adding more information on how this study is different and how it might address problems in enzyme based biosystems for chromium removal will add more value to the manuscript.
- I would like to suggest keeping the results and discussions section separately. Right now there is lot of information in the results and discussion section and it is difficult to track where the authors presenting their results and where they are trying to discuss their results with other studies. It will help readability.
Author Response
1.Thanks reviewer’s inspiring suggestions. The additional information has been supplemented in the study to address the significance in chromium removal by the system (Line 69-71 and Line 86-88), and illustrated as follow.`
Thus, the feasibility of chromate reductase in removing Cr(VI) is considered because it functions without the requirement of high organics concentration, and complicated biochemistry processes like bacterial cells but possesses high substrate specificity for Cr(VI) reduction.
Although the basic capabilities of CRB and related enzymes in removing Cr(VI) have been examined, little is known about their differences in engineering applications, such as operating conditions, operating guidelines, applicable targets, and cost analysis.
2. In this study, the discussions and results comparisons were immediately located after the results. The changes of manuscript structure may cause in difficult to track for other reviewers. Still, thanks for the reviewer’s inspiring suggestions, the separational format should be adopted in the future study.
Reviewer 2 Report
The reviewed manuscript concern operational characteristics of immobilized ochrobactrum sp. CUST210-1 biosystem and immobilized chromate reductase biosystem in continuously treating actual chromium-containing wastewater. The manuscript is well written and requires minor corrections before publication.
In details:
An abbreviation list should be added.
In the subsection "Apparatus for Continuous Cr (VI) Removal" please add the scheme of the reactor used.
In line 390 there is an incorrect notation of the sulfate anion.
Table 3 is unsightly, please correct it.
Author Response
1.According to the format of the applied sciences, abbreviations should be defined in parentheses the first time they appear in the abstract, main text, and in figure or table captions and used consistently thereafter; thus, the present format will be kept in accordance with the journal's specifications.
2.The scheme of the reactor used in the subsection "Apparatus for Continuous Cr (VI) Removal" has been supplemented. (Figure 1, page 5).
3.The notation of the sulfate anion “SO4=” has been altered to “SO42−” (Line 404).
4.We have corrected Table 3 in the original manuscript (now Table 4) by splitting a table into two separate tables.
Reviewer 3 Report
The subject of this manuscript fits to the scope of the journal and it is interest, mainly for those companies and researchers related to bioremediation and removal of heavy metals.
However, the manuscript has serious flaws: materials and methods poorly described, there are no images sustaining data from protein purification or gene expression. The structure of the protein is more like a model instead the real structure coming from crystallography based studies (and all the details about how the structure was solved are missing). In general, although interesting subject, the work is really far from the standards.
Comments have been embedded through the manuscript in order to help the authors.

Author Response
1.Information to identify the strain (Line 97-100) has been supplemented and described as follows.
To identify the Cr(VI)-reducing bacterium, the cells were first lysed, and the DNA was then extracted. The 16S rRNA gene sequence of the isolate was compared with the NCBI database using BLASTN, and the closest match to the bacterial isolate was retrieved.
2.The geographical location of this electroplating factory has bee supplemented. (Line 95)
The Ochrobactrum sp. CUST210-1 was isolated from soil in the vicinity of an electroplating factory (Changhua County, Taiwan) using the spread plate method.
3.The location of shaking culture has been supplemented. (Line 97).
The Ochrobactrum sp. CUST210-1 was isolated from soil in the vicinity of an electroplating factory (Changhua County, Taiwan) using the spread plate method and then cultivated in Luria–Bertani (LB) broth supplemented with Na2Cr2O7 [(final concentration: 300 mg L−1 Cr(VI)] in 300-mL Erlenmeyer flasks on a rotary shaker (250 rpm) at 35°C under aerobic conditions.
4.The word of "was" has been altered to "were". (Line 104)
5.The pH of the culture was adjusted by the addition of 0.1 N HCl or NaOH to the culture. (Line 110-111)
6.The information about ultrasonic processor has been supplemented. (Line 125)
After incubating the CUST210-1 strain at 35°C for 24 h, the cells were collected by centrifugation at 8,000 rpm for 15 min. The bacterial pellets were washed with 20 mM PB, and the suspension was placed in PB solution for vibration at 4°C by the ultrasonic processor (20 KHz, 130 watt).
7.The specification of the column and the main buffer solution has been provided. (Line 127-131)
To purify the chromate reductase, the supernatant was mixed with 20 mM PB, active components were separated using a HiPrep Q XL 16/10 column (16 mm × 100 mm) (GE Healthcare, Piscataway, NJ, USA), and raw chromate reductase was collected using a HiTrap Phenyl Sepharose column (7 mm × 25 mm) (GE Healthcare) and then purified using a Mono Q HR 5/5 column (5 mm × 50 mm) (GE Healthcare).
8.The chromate reductase has been purified utilizing a Mono Q HR 5/5 column, and illustrated in Line 130-131.
9.The sequence of the primers has been supplemented in Table 1. (page 4)
10.The protocol of immobilization for the enzyme was provided in Line 169-173, and described as follows.
Calcium alginate beads are made through external gel formation [32]. Three percent (w/v) sodium alginate in 50 mM sodium PB (pH 7.0) was mixed with 5% chromate reductase solution at the ratio of 1:1. The mixture was taken into a syringe and dripped directly into a gently stirred 1.5% CaCl2 solution from a height of approximately 2 cm; this resulted in formation of spherical beads of enzyme-entrapped calcium alginate.
11.The influent pH was adjusted to desired working pH level by the addition of 1 N HCl or NaOH. (Line 213-214)
12.The word of "expressed" has been altered to "found". (Line 270)
The chromate reductase of Ochrobactrum sp. CUST210-1 was found at approximately 25 kDa on the SDS-PAGE gel, suggesting that the chromate reductase was ChrR protein; similar results were reported for S. maltophilia and Alishewanella sp. induced in the presence of Cr(VI).
13.In this manuscript, we focused on the engineering applications of Ochrobactrum sp. CUST210-1 and chromate reductase in removing Cr(VI). Detailed genomic, molecular and proteomic data are pertaining to the field of basic science research. The SDS-Page picture will be provided in the relevant study.
14.In this manuscript, we focused on the engineering applications of Ochrobactrum sp. CUST210-1 and chromate reductase in removing Cr(VI). Detailed genomic, molecular and proteomic data are pertaining to the field of basic science research. The images from the gels of PCR product will be provided in the relevant study.
15.In this manuscript, we focused on the engineering applications of Ochrobactrum sp. CUST210-1 and chromate reductase in removing Cr(VI). Detailed genomic, molecular and proteomic data are pertaining to the field of basic science research. Thus, the basic crystal structure of ChrR in Ochrobactrum sp. CUST210-1 has been provided in this study, however, deep discussion of all the parameters related to the analysis will be provided in the relevant study.
Round 2
Reviewer 3 Report
Thank you very much for the time and effort addressing the comments made by this reviewer. English language and style are in general fine but minor spell/grammar check is still required. Besides, the separation and Purification of Chromate Reductase section is still uncompleted. Please, add details about the buffers used as well as the flow rate in each chromatographic step.
Author Response
1.This manuscript has been edited by Wallace Academic Editing for English language, style, spell and grammar.
2.Details about the buffers used as well as the flow rate in each chromatographic step have been provided in the separation and Purification of Chromate Reductase section as suggested by reviewer. (Line 127-132)
To purify the chromate reductase, the supernatant was mixed with 20 mM PB, active components were separated using a HiPrep Q XL 16/10 column (16 mm × 100 mm; 20 mM PB; 2 mL min-1) (GE Healthcare, Piscataway, NJ, USA), and raw chromate reductase was collected using a HiTrap Phenyl Sepharose column (7 mm × 25 mm; 50 mM sodium phosphate and 1.0 M ammonium sulphate; 1 mL min-1) (GE Healthcare) and then purified using a Mono Q HR 5/5 column (5 mm × 50 mm; 20 mM Bis-Tris; 0.5 mL min-1) (GE Healthcare).